# `CoDrug`: Conformal Drug Property Prediction with Density Estimation under Covariate Shift

**Siddhartha Laghuvarapu**
Department of Computer Science
University of Illinois Urbana-Champaign
Urbana, IL 61801
*sl160@illinois.edu*

**Zhen Lin**
Department of Computer Science
University of Illinois Urbana-Champaign
Urbana, IL 61801
*zhenlin4@illinois.edu*

**Jimeng Sun**
Department of Computer Science
Carle Illinois College of Medicine
University of Illinois Urbana-Champaign
Urbana, IL 61801
*jimeng@illinois.edu*

## Abstract

In drug discovery, it is vital to confirm the predictions of pharmaceutical properties from computational models using costly wet-lab experiments. Hence, obtaining reliable uncertainty estimates is crucial for prioritizing drug molecules for subsequent experimental validation. Conformal Prediction (CP) is a promising tool for creating such prediction sets for molecular properties with a coverage guarantee. However, the exchangeability assumption of CP is often challenged with covariate shift in drug discovery tasks: Most datasets contain limited labeled data, which may not be representative of the vast chemical space from which molecules are drawn. To address this limitation, we propose a method called `CoDrug` that employs an energy-based model leveraging both training data and unlabelled data, and Kernel Density Estimation (KDE) to assess the densities of a molecule set. The estimated densities are then used to weigh the molecule samples while building prediction sets and rectifying for distribution shift. In extensive experiments involving realistic distribution drifts in various small-molecule drug discovery tasks, we demonstrate the ability of `CoDrug` to provide valid prediction sets and its utility in addressing the distribution shift arising from de novo drug design models. On average, using `CoDrug` can reduce the coverage gap by over 35% when compared to conformal prediction sets not adjusted for covariate shift.

## 1 Introduction

Drug discovery is a challenging and complex task, with a high failure rate and limited understanding of the chemical and biological processes involved. These contribute to making drug discovery an extremely costly and time-consuming endeavor. Recently, advances in deep learning have aimed to reduce the cost of drug discovery by proposing AI methods for developing accurate *property prediction models* and *De Novo drug design models*:

- *Property prediction models* aim to aid the laborious and expensive stages of drug discovery by building accurate supervised learning models that take in a drug representation as input and output a target property [1, 2].

---

The code associated with the paper is available at https://github.com/siddharthal/CoDrug/

37th Conference on Neural Information Processing Systems (NeurIPS 2023).

- *De novo* drug design models, on the other hand, aim to discover new drug molecules that satisfy a set of pharmaceutical properties [3, 4, 5, 6].

With the high cost and significance of drug discovery, it is essential to have accurate and reliable uncertainty estimates in supervised learning models for property prediction. By providing set-valued or interval-valued estimates instead of solely relying on point estimates, uncertainty estimation enables more informed decision-making and reduces the risk of failures, making the drug discovery process more efficient. Conformal prediction (CP), pioneered by [7], offers a solution to uncertainty quantification for complex models like neural networks, by constructing provably valid prediction sets[1] in supervised learning models. Its application to drug property prediction has also been explored for various drug discovery tasks [8, 9, 10].

A crucial assumption in the CP framework is that the test samples are exchangeable with the holdout set used to calibrate the algorithm. In drug discovery, there is often a limited amount of available training or validation data. Furthermore, *de novo* drug design models or screening datasets sample molecules from a large chemical space, making the exchangeability assumption invalid.

In this paper, we deal with a situation where the training data originates from a distribution, $P(X)$, while the test data comes from a different distribution, $P^{test}(X)$. However, in both cases, the molecular properties, determined by the conditional distribution $P(Y|X)$, remain the same as they are governed by nature and unaffected by shifts in the input distribution, assuming testing parameters are stable. This is referred to as *covariate shift*. Although recent research in Conformal Prediction (CP) [11] suggests a method for correcting covariate shift, accurately estimating the precise level of covariate shift remains a practical challenge.

This paper proposes a novel and practical method for **Co**nformal **Drug** property prediction, dubbed as `CoDrug`, to improve coverage in the conformal prediction framework under covariate shift. We address the problem of non-exchangeability by quantifying the underlying covariate shift at test time and leverage recent advances in conformal prediction to obtain prediction sets. Further, we demonstrate applying `CoDrug` to obtain valid uncertainty estimates w.r.t. a target property on molecules sampled from de novo drug design models. We summarize our main contributions below:

- We propose a novel approach to create prediction sets for drug property prediction, dubbed as `CoDrug`. Using kernel density estimates (KDE) and recent advances in CP, `CoDrug` corrects for covariate shift at test time and creates prediction sets whose coverage rate is closer to the target.
- We show that the kernel density estimates are consistent, which means that asymptotically, the covariate shift is precisely adjusted for, and the coverage guarantee is recovered.
- We demonstrate the loss of coverage in property prediction tasks induced by two forms of distribution shifts - molecular scaffold splitting and molecular fingerprint splitting. Our experiments show that `CoDrug` effectively reduces the gap between actual and target coverage for prediction sets, with an average enhancement of up to 35% compared to the conformal prediction method without covariate shift adjustment. Additionally, in our experiments on molecules generated by de novo drug design models, we observe a 60% reduction in the coverage gap on average.

## 2   Related Work

Recently, deep learning techniques have been extensively studied for their potential in drug discovery, specifically in developing accurate predictive and generative models. This led to various architectures for predicting drug properties from SMILES/SELFIES strings [12], molecular graph representations[2, 1] and self-supervised learning [13]. Another area of research focused on building generative models to discover novel molecules using variational autoencoder [5, 6] and reinforcement learning [3, 4, 14].

Furthermore, several methods have been proposed for addressing uncertainty quantification in molecule property prediction, utilizing various Bayesian techniques [15]. Recently, conformal prediction methods have gained increasing attention for drug property prediction [8, 9, 10, 16, 17]. However, these studies primarily focus on generating efficient conformal predictors, without taking into account distribution shifts. Although several benchmarking datasets [18, 19] and methods [20] have been developed for drug property prediction under distribution shift, the problem of uncertainty quantification under distribution shift is still open.

Recent advancement in conformal prediction recovers the coverage guarantee for conformal prediction under known covariate shift [11]. [21] built upon [11] and proposed the Feedback Covariate Shift

---

[1]Prediction intervals can be viewed as prediction sets, with each interval being a subset of $\mathbb{R}$.

(FCS) method for the task of protein design. In practice, one cannot know the exact densities to measure the covariate shift. Like [21], we also leverage [11], but a key difference is that the training density is well-defined in [21] but unknown in ours, requiring us to estimate it. Additionally, our focus diverges from [21] as we concentrate on molecule property prediction rather than protein design.

## 3 Preliminaries

Reliable estimation of drug properties is crucial for identifying potential drug candidates. Many essential drug properties, such as toxicity, efficacy, drug-drug interactions etc. are formulated as classification problems. Consider a classification task, with each data point $Z = (X, Y) \in \mathbb{R}^d \times [K]$ ($[K] = \{0, 1, 2, ..., K - 1\}$). For instance, in Fig. 1(a), we seek to construct prediction sets for the problem of solubility classification. (Note that in practice, most drug discovery tasks are formulated as binary classification problems, with $K = 2$, but we present the general form of the methodologies.) While building an accurate base classifier ($f$) is important, we usually would like more than a point estimate of the solubility of the molecule, but also some "confidence level". This could be encoded in the form of a prediction set denoted as $\hat{C}(X) \subseteq [K]$.

The main goal we seek in such prediction sets is valid coverage: Given a target (e.g. 90%), we would like to construct a set-valued prediction (Fig. 1(a)) such that, if a molecule is water soluble, this prediction set will include the label "water soluble" with at least 90% probability. Formally, given $1 - \alpha \in (0, 1)$, and a new test molecule $(X_{N+1}, Y_{N+1})$, we would like $\hat{C}$ to be $1 - \alpha$ valid:

$$\mathbb{P}\{Y_{N+1} \in \hat{C}(X_{N+1})\} \geq 1 - \alpha. \tag{1}$$

Conformal Prediction(CP) framework enables us to achieve such validity in Eq. (1). We will expand the details in Section 4.2. Remarkably, the only requirement of CP is a hold-out calibration set where the base classifier $f$ is not trained on[2].

One critical assumption for typical CP methods is that the test and calibration data are i.i.d (or exchangeable) which is rarely realistic in drug discovery tasks. On the other hand, although the distribution of molecules $X$ changes from calibration to test time, the conditional distribution $Y|X$ is unlikely to change as the molecular properties are determined by nature and remain the same under similar experimental conditions. Formally, if we denote our calibration set as $\{(X_i, Y_i)\}_{i=1}^N$ and the test point as $(X_{N+1}, Y_{N+1})$, we have:

$$\forall i \in [N], (X_i, Y_i) \overset{i.i.d}{\sim} P^{cal} = P_X^{cal} \times P_{Y|X}^{cal} \tag{2}$$

$$(X_{N+1}, Y_{N+1}) \sim P^{test} = P_X^{test} \times P_{Y|X}^{cal}. \tag{3}$$

It is important to note that the test distribution $P^{test}$ maintains the same conditional distribution $P_{Y|X}^{cal}$ as the calibration distribution, a phenomenon known as *covariate shift*. This shift is prevalent in *de novo* drug design models, which require navigating a vast chemical space to pinpoint optimal molecules for a specific goal. However, in many drug discovery tasks, the datasets typically contain only a few thousand data points, representing a limited chemical space. Thus, when models trained on these smaller datasets are used on molecules drawn from the broader molecular space, they inevitably encounter covariate shift. Next we will lay out the exact details of constructing prediction sets with the presence of covariate shifts for supporting drug discovery applications.

## 4 CoDrug Method

### 4.1 Overview

In the subsequent subsections, we describe the three primary components of CoDrug. In Section 4.2, we first provide a brief overview of inductive conformal prediction, presenting a method for constructing valid prediction sets in scenarios both without and with distribution shifts, presuming oracle access to the unknown distributions $P_X^{test}$ and $P_X^{cal}$. Next, in Section 4.3, we present the details of the training aspects of the base energy-based classifier, emphasizing additional regularization using unlabeled data to enhance its capability to model varying molecule distributions. Finally, in Section 4.4, we employ kernel density estimation (KDE) on the embeddings or logits of the energy model trained in Section 4.3 to estimate the unknown distributions $P_X^{test}$ and $P_X^{cal}$, and rectify covariate shift using

---

[2]The assumption indicates that since the model $f$ was not trained on the calibration set, whatever over-fitting happens on $P_{Y|X}^{train}$, but not $P_{Y|X}^{cal}$. Roughly speaking, we assume the classifier's performance on the calibration set is similar to that on an unseen test set.

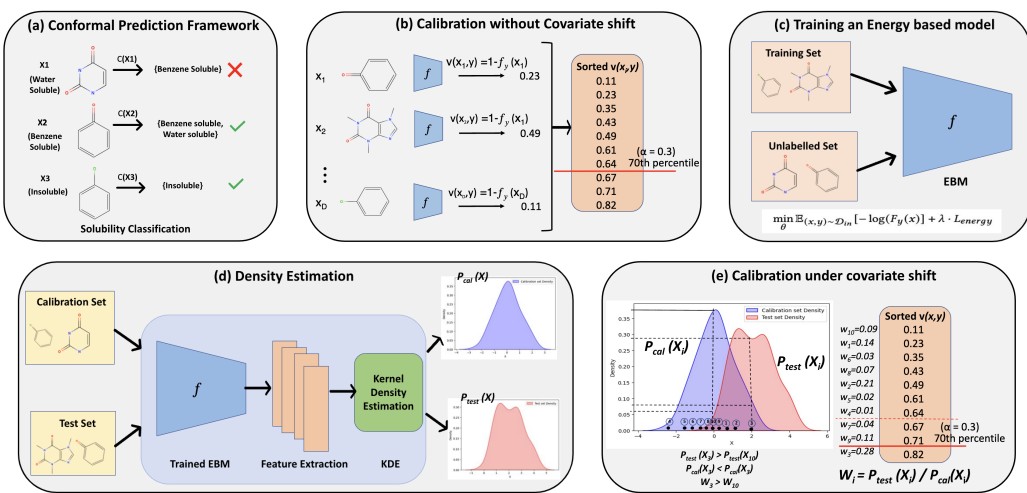

Figure 1: `CoDrug` overview: **(a)** A depiction of the conformal prediction (CP) framework. A valid prediction set includes the true label of the input molecule. **(b)** Standard procedure for computing quantiles from the calibration set when the test set is exchangeable. The calibration set's "non-conformity" scores are sorted, and the $(1-\alpha)$ quantile serves as the threshold for the conformal prediction set. **(c),(d),(e)** describe the CoDrug pipeline. **(c)** Training an Energy-based model using labeled and unlabeled data. **(d)** Density estimation: The model from (c) is used to estimate the density of the calibration and test sets. **(e)** Calibration under covariate shift: First, likelihood ratios $w_i$ are computed from the densities in (d). Then, Quantile is computed in a weighted fashion. Note how the quantile at $\alpha = 0.3$ is shifted from 0.64 (in (b)) to 0.71 to account for the distribution shift.

Section 4.2. As KDE is consistent, we regain the coverage guarantee asymptotically. Together, these elements constitute the pipeline depicted in Fig. 1.

### 4.2 Conformal Prediction Set

In this section, we will explain how to use conformal prediction to construct valid prediction sets. We will start with the case without covariate shift, and then explain how to correct for covariate shift.

#### 4.2.1 Conformal Prediction without Covariate Shift

Conformal prediction, pioneered by [7], is a powerful framework to construct prediction sets with the guarantee in Eq. (1). We assume that a base classifier $f$ is trained on a training set $\mathcal{D}_{train}$, and we have a hold-out calibration set $\mathcal{D}_{cal}$. To simplify notation, we will denote the calibration set as $\{Z_i\}_{i=1}^N$ and the test point of interest as $Z_{N+1}$. We will also abuse the notation to use $\mathcal{D}$ to denote both the empirical calibration/test set as well as the underlying distribution. Note that we ignored the training samples because they are no longer used after the classifier $f$ is trained.

We first introduce some useful definitions: (empirical) CDF and quantile function.

**The cumulative distribution function (CDF)** $F$ of a set of values $\{v_i\}_{i=1}^N$ is defined as:

$$F_{\{v_i\}_{i=1}^N} := {}^1\!/\!N \sum_{i=1}^{N} \delta_{v_i}, \text{ where } \delta_v(x) := \mathbb{1}\{x \geq v\} \tag{4}$$

**The quantile function** with respect to a CDF $F$ is:

$$Quantile(\beta; F) := \inf\{x : F(x) \geq \beta\} \tag{5}$$

Given a target coverage level $1 - \alpha \in (0, 1)$, the Mondrian inductive conformal prediction set (Mondrian ICP) is given by:

$$\hat{C}(X_{N+1}) := \{y : 1 - p_y^f(X_{N+1}) \leq t\} \tag{6}$$

$$\text{where } t := Quantile(1 - \alpha; F_{\{1-p_{Y_i}^f(X_i)\}\cup\{\infty\}}). \tag{7}$$

where, $p_y^f(x)$ corresponds to the softmax output of class $y$ from model $f$. Here, $\{v_i\}_{i=1}^N$ are defined by $v(x_i, y_i) = 1 - p_{y_i}^f(x_i)$, which are called "nonconformity scores" [22] and measure how "anomalous"

a point $z = (x, y)$ is with respect to other points from this distribution. Intuitively, we assign to each molecule a score using the same rule using $f$, which is trained on a separate data split $\mathcal{D}_{train}$. Now, we choose a threshold $t$ that is larger than $1 - \alpha$ (e.g., 90%) of the molecules. Because of our i.i.d. assumption, if we sample another molecule $Z_{N+1}$ from the same distribution, we expect its score to be lower than this threshold with a probability of $1 - \alpha$. For eg, in Fig. 1(b), notice how the threshold $t$, is computed as the value of the $Quantile$ function at $\alpha = 0.7$ ($t = 0.64$ in this case). We formally state the coverage guarantee *without covariate shift*, in the following theorem:

**Theorem 4.1.** *Assume i.i.d.* $\{(X_i, Y_i)\}_{i=1}^{N+1}$. *The $\hat{C}$ in Eq.* (6) *satisfies:*

$$\mathbb{P}\{Y_{N+1} \in \hat{C}(X_{N+1})\} \geq 1 - \alpha. \tag{8}$$

**Remarks**: Theorem 4.1 is a result of classical Mondrian inductive conformal prediction [7]. In fact, in the classification setting, instead of the i.i.d. assumption, one could make a slightly milder assumption that data are *exchangeable* within each class.

### 4.2.2 Conformal Prediction with Covariate Shift

While Theorem 4.1 provides a nice first step, the i.i.d. assumption poses a significant limitation in drug discovery. As mentioned in Eq. (3), the distributions of $X$ on $\mathcal{D}^{test}$ and $\mathcal{D}^{cal}$ can differ. In Eq. (7), we used the empirical CDF $F$ (Eq. (4)) to choose the threshold $t$. This is because of our i.i.d. assumption: a particular molecule type appears with equal probability/density in both the calibration and test sets. This is no longer the case with covariate shift, which means our $F$ needs to account for such difference in $P_X$.

Formally, recall that $P_X^{cal}$ and $P_X^{test}$ represent the density of the molecule $X$ for the calibration and test sets. We will assign a weight to each molecule $x$ that is proportional to the density/likelihood ratio $dP_X^{test}/dP_X^{cal}$ in the empirical CDF, leading to:

$$F_{x_{N+1}}^w := w(x_{N+1})\delta_\infty + \sum_{i \in [N]} w(x_i)\delta_{1 - p_{y_i}^f(x_i)}\Big/W \tag{9}$$

$$w(x') := dP_X^{test}(x')\Big/dP_X^{cal}(x'), \forall x' \tag{10}$$

$W = \sum_{i=1}^{N+1} w(x_i)$ is just a normalizing factor. The subscript $x_{N+1}$ is used to highlight that our updated CDF now depends on the test molecule $x_{N+1}$ through the weights. Here, $w(x')$ could be viewed as a likelihood ratio, and is crucial in adjusting for the covariate shift. For eg, in Fig. 1(e). Notice how the values of $w(x_i)$ depend on the densities $P_X^{cal}$ and $P_X^{test}$. In the figure, the value of weighted $Quantile$ at $\alpha = 0.3$ or the threshold $t$ is shifted from 0.64 to 0.71 to account for the shift. We formally state the modified theorem from [11] that recovers the coverage guarantee under covariate shift for Mondrian ICP:

**Theorem 4.2.** *[11] Assume that $\tilde{P}_X$ is absolutely continuous with respect to $P_X$. For any $\alpha \in (0, 1)$, let $F^w$ be defined as in Eq.* (9)*, and*

$$\hat{C}(x) = \{y : 1 - p_y^f(x) \leq Quantile(1 - \alpha; F_x^w)\} \tag{11}$$

*Then,*

$$\mathbb{P}\{Y_{N+1} \in \hat{C}(X_{N+1})\} \geq 1 - \alpha. \tag{12}$$

However, in practice, both $P_X^{test}$ and $P_X^{cal}$ are unknown, rendering Theorem 4.2 impractical. In section 4.4, we will provide a viable way to estimate $P_X^{test}$, and recover the guarantee asymptotically (namely with large calibration and test sets). In the following section, we will delve into our training methodology, which harnesses unlabeled data to effectively model varying molecular distributions.

### 4.3 CoDrug **Training Methodology**

CoDrug handles distribution shift by proposing an energy-based model formulation [23]. The core idea behind an energy-based model is to construct a function $E$ that maps an input $x$ to a scalar value, known as energy. A collection of energy values can be transformed into a probability density function $p(x)$ through the Gibbs distribution

$$p(y|x) = e^{-E(x,y)/T}\Big/e^{-E(x)/T} \tag{13}$$

Consider a discriminative neural network $f$ used in a $K$ class classification setting. $f(x)$ maps an input $x$ into $K$ real-valued scalars, which are used to derive a conditional class-wise probability:

$$p(y|x) = e^{f_y(x)/T}\Big/\sum_{y'}^{K} e^{f_{y'}(x)/T} \tag{14}$$

where, $f_y(x)$ refers to the $y^{th}$ logit of the classifier $f(x)$. In this setting, the energy function $E(x)$ can be expressed in terms of the denominator of the softmax probabilities in Eq. (14).

$$E(x; f) = -T \cdot \log \sum_{y'}^{K} e^{f_{y'}(x)/T} \tag{15}$$

Directly using embeddings from a model trained on labeled data may not yield reliable density estimates, as the model lacks knowledge of data outside the training distribution. To overcome this, we co-train the model with unlabeled molecule data. This aids the model $f$ in effectively mapping molecules with distribution shifts to a distinct embedding space. We follow [24], and use an extra regularization term in the loss function to ensure energy separation between in-distribution and out-of-distribution data. The objective function is defined as follows:

$$\min_{\theta} \mathbb{E}_{(x,y) \sim \mathcal{D}_{in}} [-\log(p_y^f(x))] + \lambda \cdot L_{energy} \tag{16}$$

where $p_y^f(x)$ refers to the softmax outputs of the classification model $f$ for class $y$, and $\mathcal{D}_{in}$ is the in-distribution training data for which labels are available. The training objective is combined with an additional term $L_{energy}$ given by:

$$L_{energy} = \mathbb{E}_{(x_{in},y) \sim \mathcal{D}_{in}} (\max(0, (E(x_{in}) - m_{in}))^2) + \mathbb{E}_{(x_{out},y) \sim \mathcal{D}_{out}} (\max(0, (m_{out} - E(x_{out})))^2) \tag{17}$$

where $\mathcal{D}_{out}$ refers to the subset of the unlabelled that is out-of-distribution (OOD). The objective of this term is to enforce a margin of separation between the training samples and the OOD data using the hyper-parameters $m_{in}, m_{out}$. Particularly, one term penalizes the model if the energy values for in-distribution data are higher than a certain value, while the other term penalizes if the OOD samples have an energy lower than a certain value. In the next section, we will explain how to use either the latent embedding of $f$ or the logits to estimate the density and correct for covariate shift.

## 4.4 Density Estimation

As discussed earlier, we need to estimate $P_X^{test}$ and $P_X^{cal}$ to correct for the covariate shift. We resort to Kernel Density Estimation (KDE), a classical nonparametric method, to estimate the density of arbitrary distributions of molecules. For a set of data $X_1, \ldots, X_n \overset{i.i.d}{\sim} \mathcal{D}$, KDE is given by:

$$\hat{p}_h(x; \mathcal{D}) = (nh)^{-1} \sum_{i=1}^{n} K(x - x_i/h), \tag{18}$$

where $K$ is a fixed non-negative *kernel* function, and $h > 0$ is a smoothing *bandwidth*. Such KDE estimates have nice asymptotic convergence properties, as formally stated in the following theorem:

**Theorem 4.3.** *[25] Assume that the true density $p$ is square-integrable and twice differentiable and that its second-order partial derivatives are bounded, continuous, and square-integrable. If $K$ is spherically symmetric on $\mathbb{R}^d$, with a finite second moment, and we choose the bandwidth $h$ such that*

$$\lim_{m \to \infty} h^d m \to \infty \text{ and } \lim_{m \to \infty} h \to 0 \tag{19}$$

*then as $m \to \infty$,*

$$\|\hat{p}_h(x) - p(x)\|_2 \overset{P}{\to} 0, \tag{20}$$

*where $\overset{P}{\to}$ means convergence in probability.*

Note that commonly used kernels, such as Gaussian kernel, satisfy the requirements.

Since $x$ here refers to molecular entities (e.g. SMILES strings), we cannot use a Gaussian kernel directly. Instead, we use embeddings or prediction logits produced by a trained model $f$ as the input to the kernel. Under the assumption that KDE accurately reflects the true density of the underlying distribution, we could construct kernel density estimators for both the calibration set and test sets (remember that we do not have access to the test labels but have access to the input $X$), and use

$$\hat{w}(x) := \hat{p}_{h_{test}}(x; \mathcal{D}_{test}) / \hat{p}_{h_{cal}}(x; \mathcal{D}_{cal}) \tag{21}$$

to replace the unknown $w$ in Eq. (10), giving us the final prediction set:

$$\hat{C}^{\texttt{CoDrug}}(x) = \{y : 1 - p_y^f(x) \leq Quantile(1 - \alpha; F_x^{\hat{w}})\} \qquad (22)$$

$$F_{x_{N+1}}^{\hat{w}} := \hat{w}(x_{N+1})\delta_\infty + \textstyle\sum_{i \in [N]} \hat{w}(x_i)\delta_{1 - p_{y_i}^f(x_i)} / \hat{W} \qquad (23)$$

where $\hat{W} = \sum_{i=1}^{N+1} \hat{w}(x_i)$ is a normalizing factor. Here, $\hat{p}_{h_{test}}(\cdot; \mathcal{D}_{test})$ is constructed using samples from the test data with an optimal bandwidth $h_{test}$ chosen on the test data via cross-validation, and $\hat{p}_{h_{cal}}(\cdot; \mathcal{D}_{cal})$ is constructed similarly but on the calibration data. It is clear that, as the number of samples from $\mathcal{D}_{cal}$ and $\mathcal{D}_{test}$ increases, $\hat{w}$ converges to $w$ in Eq. (10), and $\hat{W}^{\texttt{CoDrug}}$ recovers the coverage guarantee asymptotically. In practice, recovering asymptotic coverage on a finite amount of data is challenging. However, the coverage tends to approach the target value as we observe in our experiments. The overall procedure for density estimation is depicted in Fig. 1(d).

Algorithm 1 summarizes all the components in Section 4. In Section 5, we will verify the efficacy of `CoDrug` in property prediction tasks, and molecules sampled from de novo drug design models.

---

**Algorithm 1** Procedure for Property Prediction

**Training**:

    Split the dataset into training set $\mathcal{D}_{train}$ and calibration set $\mathcal{D}_{cal} = \{z_i\}_{i=1}^N$.
    Train a neural net classifier $f$ on $\mathcal{D}_{train}$ by minimizing Eq. (16).
    Compute the KDE $\hat{p}_{h_{cal}}(\cdot; \mathcal{D}_{cal})$ for all points in $\mathcal{D}_{cal}$ using Eq. (18).

**Test Time**, for a test set $\mathcal{D}_{test}$:

    Compute KDE $\hat{p}_{h_{test}}(\cdot; \mathcal{D}_{test})$ for all points in $\mathcal{D}_{cal}$ using Eq. (18).
    For any $x_{N+1} \in \mathcal{D}_{test}$, compute $\hat{w}(x)$ and $\hat{w}(x_i)$ for $x_i \in \mathcal{D}_{cal}$.
    Construct the prediction set $\hat{C}^{\texttt{CoDrug}}(x_{N+1})$ using Eq. (22).

---

## 5 Experiments

In this section, we put our proposed method, `CoDrug`, to the test on various drug discovery tasks. Section 5.1 describes the datasets used and key implementation details. Section 5.2.1 empirically demonstrates the loss of validity in conformal prediction sets on different drug discovery datasets. Section 5.2.2 shows how the setup improves the validity of the conformal prediction sets. Section 5.3 confirms the utility of `CoDrug` in de novo drug design. We include additional details on implementation, datasets, and hyperparameters in the appendix.

### 5.1 Data and Implementation Details

- **Splitting Strategies:** To demonstrate the effectiveness of `CoDrug` under covariate shift, we use two different strategies when creating calibration/test splits. In both strategies, we try to create calibration and test splits that are dissimilar to each other, which is a challenging but realistic setting in drug discovery. We used the DeepChem[26] library for splitting. In **scaffold splitting**, the dataset is grouped based on chemical scaffolds, representing core structures of molecules. The test set and train set consist of different scaffolds. In **fingerprint splitting**, the dataset is partitioned based on Tanimoto similarity of molecular fingerprints [27]. Molecules with the highest dissimilarity in terms of Tanimoto similarity are included in the test set.
- **Datasets:** We use four binary classification datasets for toxicity prediction (AMES, Tox21, ClinTox) and activity prediction (HIV activity), obtained from TDC [28]. To train the Energy based model, we obtained the unlabelled data from the ZINC-250k dataset [29], a subset of the ZINC that covers a large chemical space. For each dataset and split type, we removed the molecules that are similar to the training (and calibration) set from the unlabelled dataset.
- **Classification Model:** The architecture of our classifier $f$ is AttentiveFP [1], a graph neural network-based model. We chose AttentiveFP as it has state-of-the-art results in several drug property prediction tasks. It is trained using the objective function described in Eq. (16).
- **De Novo Drug Design Experiments:** In Section 5.3, we perform experiments to construct conformal prediction sets on molecules sampled from de novo drug design models. As generative models, we use REINVENT[14] and GraphGA[30] (top-ranked methods in MolOpt [31] benchmark). The models are optimized to sample molecules w.r.t. three popularly used computational oracles - QED (quantitative estimate of drug-likeness), JNK3 activity, and GSK3B activity. For building the conformal prediction sets, we chose logP as our target property, assigning values in the range of

[1.0,4.0] a class of Y=1, and Y=0 otherwise (representing the drug-like range [32]). We obtain the computational oracles from TDC [33], and generative models from MolOpt package[31].

## 5.2   Property Prediction Results

### 5.2.1   Unweighted conformal prediction (baseline)

In this section, we demonstrate the unpredictable behavior of the unweighted CP method without proper correction under distribution shift. Table 1 shows the results of conformal prediction under various distribution shift conditions. "Random" refers to the ideal/unrealistic scenario where the test and calibration samples are split randomly (aka. no distribution shift). "Scaffold" and "Fingerprint" denote scenarios in which there is a distribution shift between the test and training data outlined in the Methods section. In all scenarios, 15% training set is held out for calibration, and prediction sets are calculated using the algorithm described in Algorithm 1 without any correction.

From Table 1, we observe that the Random configuration demonstrates little loss in coverage and coverage decreases under distribution shifts (Scaffold and Fingerprint). But for fingerprint and scaffold split, unweighted CP failed to provide target coverage and exhibit unpredictable behavior. For instance, at $\alpha = 0.2$, under fingerprint split, Unweighted has a coverage of 0.34 against a target coverage of 0.8 for the AMES dataset, while achieving a very different coverage of 0.77 with scaffold split on the same dataset.

| Dataset | Random | Fingerprint | Scaffold | Random | Fingerprint | Scaffold |
|---|---|---|---|---|---|---|
|  | $\alpha$=0.1 | | | $\alpha$=0.2 | | |
| AMES(Y=0) | **0.94** | 1.00 | 0.82 | **0.85** | 1.00 | 0.66 |
| AMES(Y=1) | **0.87** | 0.63 | 0.85 | 0.78 | 0.34 | 0.77 |
| ClinTox(Y=0) | **0.88** | 0.78 | 0.84 | **0.77** | 0.58 | 0.75 |
| ClinTox(Y=1) | 0.82 | 0.80 | 0.97 | **0.78** | 0.73 | **0.81** |
| HIV(Y=0) | **0.90** | **0.93** | **0.91** | **0.80** | 0.89 | **0.81** |
| HIV(Y=1) | **0.89** | **0.84** | **0.87** | **0.80** | **0.72** | 0.73 |
| Tox21(Y=0) | **0.90** | 0.77 | **0.89** | **0.80** | 0.65 | **0.75** |
| Tox21(Y=1) | **0.86** | 0.97 | **0.93** | 0.72 | 0.97 | **0.82** |

Table 1: Unweighted CP's (baseline) coverage under various distribution shifts (or absence thereof) should ideally align closely with the target $1 - \alpha$. However, in most datasets with fingerprint and scaffold splits—reflective of more realistic scenarios—the baseline method falls short. Often, substantial deviations in coverage confirm the unpredictability of unweighted CP when exchangeability ceases to apply. Here, values not significantly deviating from $1 - \alpha$ at a p-value of 0.05 are highlighted in bold, indicating desirable performance.

### 5.2.2   Weighted conformal prediction improves coverage

In Table 2, we present the benefits of using weighted CP via `CoDrug`. The table depicts results from conformal prediction using 3 different schemes.

- `CoDrug` **(Energy):** This variant of `CoDrug` uses weights computed from KDE on the prediction logits of the trained EBM, as described in Section 4.3.
- `CoDrug` **(Feature):** This variant of `CoDrug` builds the KDE instead on the features extracted from the penultimate layer of the trained EBM.
- **Unweighted:** Refers to the unweighted prediction conformal prediction (baseline).

In both weighting schemes of `CoDrug`, we use KDE to estimate densities and find that weighting using energies improves the coverage towards the target coverage $1 - \alpha$ in most cases. We notice the highest improvement in the Fingerprint splitting scenario for the AMES(Y=1) category, where the coverage improved from 0.63 to 0.88 (target coverage 0.9). Note that the coverage is "improved" if it is closer to $1 - \alpha$ - improvement does not always mean higher coverage, because an unusually high coverage often indicates unpredictable behavior of the underlying model.

While our energy-weighting approach generally improves coverage, there are rare instances where it underperforms compared to the baseline. A prominent example is the ClinTox dataset with $Y = 1$, which sees limited improvement or even a reduction in coverage. This is due to the constraints of the density estimation procedure, which relies on the quantity of available data. Notably, this dataset is the smallest and most imbalanced, with only 19 points in the calibration set for class $Y = 1$.

Additionally, our results show that using energy weighting leads to better overall coverage than directly weighting the features. This is likely because the energy values are two-dimensional, while the features are an eight-dimensional vector: As the dimension of the feature input to KDE increases, one typically requires more samples to get a high-quality density estimate.

| Dataset | Fingerprint Splitting | | | Scaffold Splitting | | |
|---|---|---|---|---|---|---|
| | CoDrug (Energy) | CoDrug (Feature) | Unweighted (baseline) | CoDrug (Energy) | CoDrug (Feature) | Unweighted (baseline) |
| AMES(Y=0) | **0.93(0.03)** | **0.87(0.03)** | 1.00(0.00) | 0.85(0.02) | **0.89(0.02)** | 0.82(0.01) |
| AMES(Y=1) | 0.88(0.03) | **0.90(0.03)** | 0.63(0.05) | 0.83(0.01) | 0.79(0.01) | **0.85(0.03)** |
| ClinTox(Y=0) | **0.86(0.04)** | 0.76(0.02) | 0.78(0.02) | **0.90(0.03)** | 0.83(0.01) | 0.84(0.00) |
| ClinTox(Y=1) | 0.73(0.00) | 0.69(0.08) | **0.80(0.00)** | **0.85(0.03)** | 0.83(0.00) | 0.97(0.04) |
| HIV(Y=0) | **0.89(0.06)** | 0.87(0.07) | 0.93(0.04) | 0.82(0.08) | 0.82(0.04) | **0.91(0.01)** |
| HIV(Y=1) | **0.92(0.05)** | 0.95(0.03) | 0.84(0.07) | **0.90(0.01)** | 0.90(0.05) | 0.87(0.03) |
| Tox21(Y=0) | **0.90(0.02)** | 0.80(0.02) | 0.77(0.03) | **0.91(0.03)** | 0.83(0.05) | 0.89(0.05) |
| Tox21(Y=1) | 0.97(0.00) | **0.96(0.01)** | 0.97(0.00) | 0.86(0.05) | **0.91(0.05)** | 0.93(0.03) |

Table 2: Coverage of `CoDrug` and baseline unweighted CP, under different datasets and distribution shifts at $\alpha = 0.1$. The realized coverage rate closest to the target coverage $1 - \alpha$ (best) is marked in **bold**. The second best coverage (in case better than unweighted) is marked in **bold and gray**. Results are averaged over 5 random runs. Results for different $\alpha$ values are available in appendix.

### 5.2.3 Ablation studies

In this section, we present an analysis of the importance of various components in the CoDrug pipeline - KDE, energy regularization term ( Eq. (16)), and covariate shift correction. In addition to `CoDrug (Energy)` and Unweighted (baseline) reported in the previous section, we also compare with:

- `CoDrug (NoEnergy)`: We use the same protocol as `CoDrug (Energy)` but the models are trained without the energy regularization term $\mathcal{L}_{energy}$ in Eq. (16).
- Logistic (Energy): In this experiment, the features are same as `CoDrug (Energy)`, but KDE is not used to estimate densities. Instead, the weights $w(x_i)$ in Eq. (10), are given by $\hat{p}_{(x_i)}/1-\hat{p}_{(x_i)}$, where $\hat{p}_{(x_i)}$ is obtained by fitting a classifier to features in calibration and test sets (suggested by [11]).

The results are depicted in Table 3. In the table, we compile the "mean absolute coverage deviation" across all the datasets and different random runs from all the experiments reported in Table 2 at different values of $\alpha$ (i.e Mean of $|\text{Observed\_Coverage} - (1 - \alpha)|$ across the experimental runs). The results reveal that `CoDrug (Energy)`, the proposed method, is closest to the target coverage in almost all different values of $\alpha$. We paid close attention to the "Tail 25%", where we presented the metrics for the worst 25% performing experiments and `CoDrug (Energy)` outperforms all the other variants in comparison by a substantial margin inducting that all the different components in the CoDrug pipeline are helpful. The mean absolute coverage deviation from the target at $\alpha = 0.1$ for `CoDrug (Energy)` is 0.052, a relative improvement of about 35% over that of Unweighted (0.081).

| Method | Mean absolute coverage deviation | | | Mean absolute coverage deviation (tail 25%) | | |
|---|---|---|---|---|---|---|
| | $\alpha = 0.3$ | $\alpha = 0.2$ | $\alpha = 0.1$ | $\alpha = 0.3$ | $\alpha = 0.2$ | $\alpha = 0.1$ |
| Unweighted (baseline) | 0.157 (0.14) | 0.12 (0.12) | 0.081 (0.07) | 0.347 (0.14) | 0.276 (0.13) | 0.176 (0.08) |
| Logistic(Energy) | 0.123 (0.13) | 0.106 (0.13) | 0.083 (0.14) | 0.315 (0.11) | 0.263 (0.17) | 0.222 (0.22) |
| CoDrug (NoEnergy) | 0.112 (0.11) | 0.083 (0.09) | **0.047 (0.05)** | 0.288 (0.05) | 0.215 (0.08) | 0.112 (0.03) |
| CoDrug (Energy) | **0.104 (0.09)** | **0.079 (0.07)** | 0.052 (0.05) | **0.233 (0.05)** | **0.179 (0.07)** | **0.11 (0.04)** |

Table 3: Ablations: Results comparing various versions of the proposed framework. At each $\alpha$, the mean of deviations from target coverage across all the experiments and random seeds is computed (Smaller is better). `CoDrug (Energy)` has the least deviation from coverage and a substantial difference when only the worst performing 25% of the experiments are considered.

### 5.3 Application in de novo drug design.

In this section, we examine `CoDrug`'s application in de novo drug design models, which navigate a large chemical space to find optimized molecules using a computational oracle. After molecule sampling, validating their experimental properties, such as ADMET (Absorption, Distribution, Metabolism, Excretion, Toxicity), is crucial for safety and efficacy. When a machine learning model trained on such properties is available, assessing the uncertainty associated with the predictions before experimental validation is critical. However, note that the distribution of sampled molecules may substantially deviate from the training data, affecting the prediction sets' target coverage from CP.

In this section, we demonstrate the application of `CoDrug` on molecules generated by a de novo drug design model. We consider the de novo drug design model as a black box, that has been optimized w.r.t a certain objective. We experiment with two models - GraphGA [30] and Reinvent [14]. To predict properties, we compiled a dataset of logP values, as it can be computed cheaply with a computational oracle. We note that in reality, this dataset could correspond to experimental properties like ADMET. However, since it is not feasible to validate these properties for molecules generated from de novo drug design models, we use logP to demonstrate the method. The results of our experiments are depicted in Table 4 at a target alpha value of 0.1. Our proposed method consistently enhances coverage in all cases and exhibits a substantial improvement over the unweighted version. For example, in the "gsk3b+qed" objective, the unweighted version has a coverage of 0.44 against a target of 0.9, whereas our proposed method improves coverage substantially. The mean absolute coverage deviation from the target at $\alpha = 0.1$ for `CoDrug` (Energy) is 0.05, a relative improvement of over 60% on the Unweighted version (0.14).

| | | REINVENT | | GraphGA | |
|---|---|---|---|---|---|
| Objective | Y | CoDrug (Energy) | Unweighted | CoDrug (Energy) | Unweighted |
| JNK3+QED | 0 | **0.95 (0.01)** | 0.62 (0.12) | **0.86 (0.0)** | 0.84 (0.01) |
| JNK3+QED | 1 | **0.91 (0.01)** | 0.99 (0.0) | 0.93 (0.01) | **0.89 (0.02)** |
| GSK3b+QED | 0 | **0.81 (0.04)** | 0.44 (0.16) | **0.87 (0.0)** | 0.75 (0.01) |
| GSK3b+QED | 1 | 0.79 (0.08) | **1.0 (0.0)** | **0.98 (0.0)** | 1.0 (0.0) |
| QED | 0 | **0.96 (0.0)** | 0.83 (0.04) | **0.96 (0.0)** | 0.69 (0.1) |
| QED | 1 | **0.92 (0.01)** | 0.84 (0.05) | **0.98 (0.0)** | 0.99 (0.0) |

Table 4: Observed coverages on molecules sampled by generative models at $\alpha = 0.1$. The realized coverage rate closest to the target coverage$(1 - \alpha)$ are marked in bold. For each experiment, a set of 200 points optimized w.r.t. the "Objective" using the generative models GraphGA and REINVENT are sampled. The target property for prediction is logP (1.0 < logP < 4.0 is considered Y=1; Y=0 otherwise [32]). Using the proposed method improves coverage in almost all scenarios.

## 6  Limitations

While we demonstrated that KDE can provide asymptotic coverage guarantees, this may not necessarily translate to improved performance in scenarios with limited sample sizes. In our experiments, we do acknowledge that there are a few instances where the improvement in coverage is limited, where the availability of data is limited. As such, a direction for further research is to explore ways to obtain likelihood ratios that are more data-efficient, particularly in scenarios with smaller calibration sets.

It is worth noting that our current work focuses on addressing the coverage gap in classification tasks, and regression tasks were not included in this study. However, we recognize the importance of uncertainty quantification in regression settings, especially for various critical drug properties represented as regression problems, where our proposed framework can be extended with modifications. Furthermore, our current work primarily focuses on small molecules, yet covariate shift is a common phenomenon in various chemical and biological contexts. While this means that our framework could be more generally applied, obtaining high-quality feature vectors for computing likelihood in different applications remains a challenge that warrants further research.

## 7  Conclusion

We present a new method for uncertainty quantification in drug discovery, `CoDrug`, that effectively addresses the problem of co-variate shifts in test data. The proposed method involves a combination of three key steps, training an energy-based model for feature extraction and base classification, performing density estimation using KDE, and use the KDE to correct for covariate shift in conformal prediction to recover valid coverage. The results obtained in this study demonstrate the effectiveness of `CoDrug` in predicting valid conformal prediction sets and its utility in de novo drug design experiments. Our current work is limited to classification tasks in small molecules, but exploring its application to other chemical and biological tasks with covariate shifts is interesting for future work, including adapting the framework for regression tasks.

## Acknowledgments and Disclosure of Funding

This work was supported by NSF award SCH-2205289, SCH-2014438, and IIS-2034479. This project has been funded by the Jump ARCHES endowment through the Health Care Engineering Systems Center.

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

# A Proofs

## A.1 Proofs for Theorem 4.1

Denote $T_i = 1 - p_{Y_i}^f(X_i)$ as a new random variable. Because of our i.i.d. assumption (and that $f$ is not trained on the calibration set), $T_1, \ldots, T_{N+1}$ are also i.i.d., which means that for $m = 0, 1, \ldots, N$:

$$\mathbb{P}\{|\{i \in [N] : T_{N+1} > T_i\}| = m\} \leq \frac{1}{N+1} \tag{24}$$

$$\implies \mathbb{P}\{|\{i \in [N] : T_{N+1} > T_i\}| \geq (N-m)\} \leq \frac{m+1}{N+1} \tag{25}$$

The left-hand-side probability is $\leq$ (instead of $<$) the right-hand-side due to the case when $p_y^f$ is not continuous. Note also that

$$1 - p_{Y_{N+1}}^f(X_{N+1}) > t \implies |\{i \in [N] : T_{N+1} > T_i\}| \geq \lceil (1-\alpha)(N+1) \rceil \tag{26}$$

which means

$$\mathbb{P}\{1 - p_{Y_{N+1}}^f(X_{N+1}) > t\} \leq \frac{\lfloor \alpha(N+1) \rfloor}{N+1} \leq \alpha. \tag{27}$$

Finally, note that

$$Y_{N+1} \notin \hat{C}(X_{N+1}) \implies 1 - p_{Y_{N+1}}^f(X_{N+1}) > t, \tag{28}$$

we thus have

$$\mathbb{P}\{Y_{N+1} \in \hat{C}(X_{N+1})\} \geq 1 - \alpha. \tag{29}$$

$\square$

## A.2 Proofs for Theorem 4.2

This is a direct result of Corollary 1 in [11], with our nonconformity score $1 - p_Y^f(X)$ plugged in.

# B Data and implementation details

## B.1 Training Details of the base classifier:

- **Deep Learning Frameworks:** We use the Pytorch framework for the implementation of the models. The Graph Neural Network backbone is obtained from the DGL-LifeSci library [34].
- **Training hyperparameters:** We train the model using the PyTorch Lightning Framework for training. We use the ADAM Optimizer [35]. The batch size is set to 64, and the learning rate is set to 0.001.
- **Architecture Details:** The model architecture consists of a GNN layer (Attentive FP [1]), , a readout layer, 2 hidden FCNN layers, and an output layer. The hidden state size in GNN is set to 512 dimensions. The linear layers have 256, and 8 dimensions respectively.
- **Cheminformatics Processing :** We use the RDKit library for handling molecular entities in Python. We use the DeepChem library for generating dataset splits.
- **Energy Regularization hyperparameters :** The parameters $m_{in}$ and $m_{out}$ in Eq. (15) are set to -5 and -35 respectively, and the parameter $\lambda$ in Eq. (16) is set to 0.01. All of these hyperparameters are obtained from the reference implementation in [24].
- **Splitting Ratio:** The datasets are split in the ratio of 70:15:15, for training, calibration and testing the CP model.
- **Error bars:** All the experiments reported are over 5 random runs. The mean and the standard deviation across the random runs is reported in the table.

## B.2 Splitting strategies:

In this section, we discuss the two strategies employed for creating calibration and test sets to ensure their dissimilarity. The implementation of these strategies was based on the DeepChem library [26]. We provide detailed explanations of the algorithms used for splitting below.

- **Scaffold Splitting:** The core structure of a molecule is represented by scaffolds [36]. Scaffold splitting aims to generate train and test sets that do not share any common scaffolds, thereby creating a challenging yet realistic scenario of distribution shift. This strategy is commonly used to evaluate out-of-distribution prediction algorithms [20, 18]. The scaffold splitting procedure is as follows:
  - First, the scaffolds of all molecules in the datasets are identified.
  - The scaffolds are sorted by their frequency in the dataset.
  - The least frequent scaffolds are added to the test set until the desired number of test data points is reached.
  - The remaining points are randomly divided into training and calibration sets.
- **Fingerprint Splitting:** A molecular fingerprint[27] is a compact binary representation of a molecule's structural features, capturing important information about its chemical composition and spatial arrangement. Fingerprint splitting utilizes molecular fingerprints to create distinct train and test sets. The primary objective is to include data points in the test set that exhibit the least maximum pairwise Jaccard similarity of fingerprints. The fingerprint-splitting procedure is as follows:
  - First, the molecular fingerprints are computed for all the molecules in the dataset using Extended Connectivity Fingerprints (ECFP) [27].
  - Pairwise Jaccard similarity is calculated for each data point in the dataset, considering its similarity with all other points. The Jaccard similarity between two fingerprints is determined by dividing the size of their intersection by the size of their union.
  - The test set construction begins by selecting the data point with the least maximum Jaccard similarity to any other point in the dataset. This point is added to the test set.
  - The iterative process continues until the desired number of test data points is reached. The remaining data points are assigned to the training/calibration set.

## B.3 Property prediction datasets

### B.3.1 Labelled data

We used four commonly used classification benchmarking datasets from ADMET properties. The datasets are obtained from Therapeutics Data Commons (TDC) [28]. We include the statistics of all the datasets used for property prediction in Table 5.

- **AMES Mutagenicity**: Mutagenicity is a vital toxicity measure that measures the ability of the drug to induce genetic mutations. The dataset consists of toxicity classes for over 7000 compounds.

- **Tox21**: A data challenge that consists of qualitative toxicity measurements on 12 different targets. For the scope of this work, we picked the largest assay in the collection with over 6000 compounds.
- **ClinTox**: A collection of compounds that includes drugs that have failed in clinical trials for toxicity reasons and the ones with successful outcomes. This dataset contains about 1500 drugs.
- **HIV Activity**: The dataset from screening results published by Drug Therapeutics Program (DTP) AIDS Antiviral Screen. It measures the ability to inhibit HIV replication for over 40,000 compounds.

| Dataset | #Positive | #Negative | #Total | Task |
|---------|-----------|-----------|--------|------|
| Tox21 | 309 | 6,956 | 7,265 | prediction |
| ClinTox | 112 | 1,366 | 1,478 | prediction |
| AMES | 3,974 | 3,304 | 7,278 | prediction |
| HIV | 1,443 | 39,684 | 41,127 | prediction |
| ZINC | 0 | 0 | 250,000 | pre-training |

Table 5: **Dataset statistics**

### B.3.2 Unlabelled data

To train the EBM, we obtained the unlabelled data from the ZINC-250k dataset [29]. This is a subset of the ZINC database, typically used for pre-training generative models, and covers a large chemical space. For each dataset and split type, we removed the molecules that are similar to the training (and calibration) set.

- In the case of scaffold splitting, we remove the molecules containing scaffolds present in the training set.
- In the case of Fingerprint splitting, we compute the Tanimoto similarity with respect to the molecules in the training set and include only those molecules with similarities less than the minimum pairwise similarity in the training set.

### B.4 Hyperparameters for Kernel Density Estimation(KDE):

Determining the bandwidth($h$), for Kernel Density Estimation (KDE) is crucial for accurate density estimation. To find the optimal value of $h$, we employ K-fold cross-validation (CV) using the scikit-learn library, with $k = 10$ folds. The following procedure is applied for each dataset:

- The dataset is divided into $k$ splits or folds.
- We fit the KDE model using a range of $h$ values, specifically choosing 25 uniformly spaced intervals from $10^{-1.3}$ to $10^1$.
- The fitted KDE models are evaluated by computing the log probability on the holdout split for each $h$ value.
- The $h$ value that yields the highest average log probability across the $k$ folds is selected as the optimal bandwidth for fitting the KDE model.

### B.5 Ablation models:

As discussed in Section 5.2.3, we perform the following ablations:

- `CoDrug` (NoEnergy): The main objective of this variant of the method is to understand the significance of training with energy regularization described in 4.3. In this study, we follow the same protocol as `CoDrug` (Energy), but during the training, we do not use the regularization term $\mathcal{L}_{energy}$ in 16.
- Logistic (Energy): Logistic (Energy): In this variation, the training procedure for the model remains the same, but instead of employing Kernel Density Estimation (KDE) for estimating molecular density, we utilize a logistic classifier. This approach, initially utilized by Tibshirani et al. (2020) [11] in their experiments, involves training the logistic classifier using the same input features as those used in KDE.
  For training the classifier, the samples in the calibration set are labeled as 0, while the samples in the test set are labeled as 1. The weight assigned to a point $x$ in the calibration set is determined by the estimate $\hat{p}(x)$ of $\mathcal{P}(C = 1|X = x)$ obtained from the classifier, and is calculated as $\hat{p}(x)/1-\hat{p}(x)$. To implement the logistic classifier, we utilized the scikit-learn library, employing the default hyperparameters for the classifier.

### B.6 Details of de novo drug design experiments:

In this section, we present details of the experiments described in Section 5.3. We first describe the de novo drug design models used for the experiments in Appendix B.6.1. We sample 200 points from these generative models on properties described in Appendix B.6.2. Our objective is to construct valid prediction sets on the sampled molecules pertaining to the property discussed in Section B.6.2.

#### B.6.1 De novo drug design models

As our main objective is to show that CoDrug is effective in estimating uncertainties on molecules sampled from de Novo drug design models, we experiment on molecules sampled from two different de novo drug design models - Reinvent and Graph GA. These two models are top-ranked on the MolOpt benchmark [31]. We used the implementation provided by MolOpt to run our experiments.

- **Reinvent:** Reinvent[14] is a reinforcement learning-based de novo drug design model. The model uses an RNN to generate SMILES strings.
- **GraphGA**: GraphGA [30] is a genetic algorithm based de novo drug design model that generates molecular graphs.

#### B.6.2 Objective functions used for optimization

In our experiments, we used molecule sets obtained from optimizing the molecules on the following properties. All the oracles are obtained from TDC [28].

- **QED:** QED stands for Quantitative Estimate of Drug-likeness. It is a computational metric used in drug discovery to assess the "drug-likeness" of a compound. QED provides a quantitative measure of how likely a molecule is to possess drug-like properties based on its chemical structure.
- **QED + JNK3 activity:** JNK3 activity refers to the activity of a molecule against a c-Jun N-terminal Kinases-3 (JNK3) protein. This is a common Oracle function used in benchmarking de novo drug design models. The oracle is built using a random forest classifier using ECFP6 fingerprints using the ExCAPE-DB dataset. In addition, we also add QED score to the Oracle output to restrict molecule search to a "drug-like" region.
- **QED + GSK3b activity**: GSK3b, which stands for glycogen synthase kinase 3 beta, is an enzyme encoded by the GSK3b gene in humans. Dysregulation and abnormal expression of GSK3b have been linked to a heightened vulnerability to bipolar disorder. Similar to previous case, the oracle is built using a random forest classifier that utilizes ECFP6 fingerprints from the ExCAPE-DB dataset. We also add QED score to the Oracle output.

#### B.6.3 Details of the property prediction experiments

As discussed in the main paper, we choose logP as our target property, i.e. the property for which we wish to obtain uncertainty estimates. LogP, also known as the logarithm of the partition coefficient, is a property used to quantify the lipophilicity of a drug molecule. The lipophilicity of a drug molecule, as determined by LogP, plays a crucial role in its pharmacokinetic properties. It is accepted that a drug-like molecule would have logP in the range of [1.0, 4.0] [32] and hence, in our experiments, we assign a label of Y=1 for logP in [1.0, 4.0]; Y=0 otherwise.

logP can be computed cheaply using a computational oracle, which obtained it from TDC[28]. Note that this property in reality would be an experimentally determined property (such as ADMET properties), but the obvious challenge in validating our method on such properties is that it is not possible to obtain ground truth values for novel molecules obtained from de novo drug design models. Nevertheless, since CP is agnostic to the underlying prediction model, we deem that the performance would remain robust across different properties and hence would potentially be beneficial in real-world drug discovery campaigns.

For the curation of the training set, we randomly pick a set of 20 scaffolds from the ZINC250k dataset [29], and pick 500 points from each scaffold (making a total of 10000 points). This is our training and calibration data. We assign labels to this set based on the above-mentioned criteria and train the model using the same procedure as other property predictors as described in section 5. Note that the test set for this exercise is the molecules sampled from de novo drug design models.

# C Results

In section Section 5, we have provided results for experiments at $\alpha = 0.1$. Here, we provide additional results for the experiments at $\alpha = 0.05$ and $\alpha = 0.2$.

## C.1 Property prediction results

| Dataset | Fingerprint split ($\alpha = 0.05$) | | | Fingerprint split ($\alpha = 0.2$) | | |
|---|---|---|---|---|---|---|
| | CoDrug(energies) | CoDrug(features) | Unweighted(baseline) | CoDrug(energies) | CoDrug(features) | Unweighted(baseline) |
| AMES(Y=0) | **0.96(0.07)** | **0.93(0.09)** | 1.00(0.01) | **0.85(0.06)** | **0.78(0.06)** | 1.00(0.00) |
| AMES(Y=1) | **0.94(0.08)** | **0.95(0.08)** | 0.78(0.19) | **0.79(0.05)** | **0.82(0.06)** | 0.34(0.14) |
| ClinTox(Y=0) | **0.94(0.08)** | 0.83(0.04) | 0.91(0.07) | **0.68(0.07)** | **0.67(0.04)** | 0.58(0.04) |
| ClinTox(Y=1) | 0.88(0.14) | 0.73(0.03) | **0.93(0.00)** | **0.73(0.00)** | 0.45(0.03) | **0.73(0.00)** |
| HIV(Y=0) | **0.94(0.10)** | **0.94(0.14)** | 0.96(0.08) | **0.81(0.08)** | **0.74(0.13)** | 0.89(0.07) |
| HIV(Y=1) | **0.95(0.08)** | **0.97(0.07)** | 0.90(0.14) | **0.85(0.06)** | 0.90(0.04) | 0.72(0.11) |
| Tox21(Y=0) | **0.93(0.05)** | **0.85(0.03)** | 0.83(0.02) | **0.87(0.03)** | **0.71(0.03)** | 0.65(0.02) |
| Tox21(Y=1) | **0.97(0.01)** | **0.97(0.02)** | 0.97(0.01) | **0.95(0.01)** | **0.93(0.02)** | 0.97(0.00) |
| Dataset | Scaffold split ($\alpha = 0.05$) | | | Scaffold split ($\alpha = 0.2$) | | |
| AMES(Y=0) | **0.92(0.03)** | **0.93(0.04)** | 0.90(0.04) | **0.73(0.03)** | **0.80(0.02)** | 0.66(0.04) |
| AMES(Y=1) | 0.90(0.02) | 0.87(0.03) | **0.91(0.02)** | 0.72(0.01) | 0.66(0.02) | **0.77(0.02)** |
| ClinTox(Y=0) | **0.95(0.05)** | 0.89(0.01) | 0.94(0.02) | **0.80(0.03)** | 0.74(0.01) | 0.75(0.01) |
| ClinTox(Y=1) | **0.97(0.05)** | **0.95(0.02)** | **0.97(0.02)** | 0.53(0.07) | 0.77(0.03) | **0.81(0.02)** |
| HIV(Y=0) | 0.89(0.05) | 0.88(0.07) | **0.96(0.01)** | 0.72(0.07) | 0.71(0.06) | **0.81(0.01)** |
| HIV(Y=1) | **0.95(0.03)** | 0.94(0.07) | 0.94(0.06) | **0.81(0.01)** | **0.81(0.05)** | 0.73(0.05) |
| Tox21(Y=0) | **0.95(0.07)** | 0.88(0.04) | 0.94(0.13) | **0.81(0.05)** | 0.73(0.05) | 0.75(0.09) |
| Tox21(Y=1) | **0.94(0.06)** | **0.96(0.05)** | **0.97(0.04)** | 0.77(0.05) | **0.80(0.04)** | 0.82(0.06) |

Table 6: Coverage of `CoDrug` and baseline unweighted CP, under different datasets and distribution shifts at $\alpha = 0.05$ and $\alpha = 0.2$. The realized coverage rate closest to the target coverage $1 - \alpha$ (best) is marked in **bold**. The second best coverage (in case better than unweighted) is marked in **bold and gray**. Results are averaged over 5 random runs.

## C.2 De Novo Drug design experiment results

| Objective | Y | REINVENT ($\alpha = 0.05$) | | GraphGA ($\alpha = 0.05$) | | REINVENT ($\alpha = 0.2$) | | GraphGA ($\alpha = 0.2$) | |
|---|---|---|---|---|---|---|---|---|---|
| | | CoDrug (Energy) | Unweighted | CoDrug (Energy) | Unweighted | CoDrug (Energy) | Unweighted | CoDrug (Energy) | Unweighted |
| QED | 0 | 0.97 (0.02) | **0.94 (0.06)** | **0.97 (0.02)** | 0.81 (0.23) | **0.87 (0.1)** | 0.53 (0.25) | **0.75 (0.09)** | 0.41 (0.26) |
| QED | 1 | 0.97 (0.02) | **0.96 (0.07)** | **1.0 (0.01)** | **1.0 (0.01)** | **0.81 (0.08)** | 0.78 (0.26) | 0.9 (0.08) | **0.88(0.13)** |
| JNK3+QED | 0 | **0.89 (0.15)** | 0.83 (0.34) | **0.94 (0.02)** | 0.93 (0.09) | **0.72 (0.27)** | 0.2 (0.4) | **0.73 (0.06)** | 0.73 (0.15) |
| JNK3+QED | 1 | **0.98 (0.01)** | 1.0 (0.0) | **0.98 (0.05)** | 0.92 (0.09) | **0.86 (0.1)** | 0.94 (0.1) | **0.86 (0.04)** | 0.64 (0.27) |
| GSK3b+QED | 0 | **0.91 (0.13)** | 0.63 (0.33) | **0.91 (0.03)** | 0.77 (0.17) | **0.71 (0.21)** | 0.39 (0.42) | **0.74 (0.05)** | 0.36 (0.18) |
| GSK3b+QED | 1 | **0.92 (0.15)** | 1.0 (0.0) | **0.98 (0.02)** | 1.0 (0.00) | **0.82 (0.37)** | 0.95 (0.04) | **0.9 (0.07)** | 0.97 (0.04) |

Table 7: Observed coverages on molecules sampled by generative models at $\alpha = 0.05$ and $\alpha = 0.2$. The realized coverage rate closest to the target coverage $(1 - \alpha)$ is marked in bold. For each experiment, a set of 200 points optimized w.r.t. the "Objective" using the generative models GraphGA and REINVENT are sampled, similar to the procedure in Appendix B.6.1. Using the proposed method improves coverage in almost all scenarios.

# D List of commonly used notations and terms

Table 8: List of key notations used in the paper

| Symbol | Description |
|---|---|
| $\alpha$ | Refers to the user-defined target coverage level in conformal prediction. It determines the confidence level of the prediction regions. |
| $\hat{C}(x_i)$ | Corresponds to a prediction set of an input $x_i$ obtained from a conformal prediction method. |
| $X_i$ | Refers to the input features corresponding to a data point $i$ in a machine learning model. |
| $Y_i$ | Refers to the label corresponding to a data point $i$ in a machine learning model. |
| $f$ | Refers to the base classifier of the model. It is the underlying algorithm or model used to make predictions on the data. |
| $f_y(x)$ | Refers to the $y^{th}$ logit of the classifier $f$ on a data point $x$ without applying the softmax layer. |
| $p_y^f(x)$ | denotes the classwise probability scores of the data point $x$ with respect to the model $f$ after the application of the softmax layer on the logits $f_y(x)$. It represents the probability of the data point belonging to class $y$. |
| $E(x)$ | Refers to the energy of a data point $x$ in an energy-based model. The energy is computed based on the logits of the classifier. |
| $F_{\{v_i\}_{i=1}^N}$ | Refers to an (unweighted) cumulative distribution function computed from a set of values $\{v_i\}_{i=1}^N$. |
| $w(x_i)$ | Refers to the likelihood ratio or weight assigned to a data point $i$ in weighted conformal prediction. It quantifies the importance of the data point during calibration. |
| $F_{x_{N+1}}^w$ | Weighted cumulative distribution function computed using the likelihood ratios $w(x_i)$ from weighted conformal prediction. It represents the distribution of the weighted values. |
| $\mathcal{D}_{train}$ | Refers to the distribution of molecules from which the training set is sampled. It represents the underlying data distribution used to train a machine-learning model. |
| $\mathcal{D}_{cal}$ | Refers to the distribution of molecules from which the calibration set is sampled. It represents the underlying data distribution used to calibrate a conformal prediction model. |
| $P_X^{cal}$ | Refers to the true density of an input molecule $X$ in the calibration set distribution. |
| $P_X^{test}$ | Refers to the true density of an input molecule $X$ in the test set distribution. |
| $\hat{p}_{h_{test}}$ | Refers to the density of an input molecule $X$ computed from kernel density estimation (KDE) on the test set distribution. It is an estimation of the probability density function of the input molecule in the test set distribution. |
| $\hat{p}_{h_{cal}}$ | Refers to the density of an input molecule $X$ computed from KDE on the calibration set distribution. It is an estimation of the probability density function of the input molecule in the calibration set distribution. |

Table 9: List of commonly used terms in the paper

| Term | Description |
|---|---|
| Activity (property) | Refers to the ability of a drug to bind to a specific target molecule and produce a biological effect. It is an important property to consider in drug discovery. |
| ADMET properties | Refers to the absorption, distribution, metabolism, excretion, and toxicity of a drug candidate. These properties play a critical role in determining the safety and efficacy of a drug. |

| | |
|---|---|
| Alpha | In conformal prediction, alpha refers to the user-defined confidence level used to construct the prediction sets. The parameter determines the amount of error that the user is willing to tolerate in the predictions and is typically set to a small value, such as 0.1 or 0.2. A smaller alpha typically results in a wider prediction set. |
| Calibration( of conformal prediction) | In the Mondrian Inductive Conformal Prediction framework, calibration refers to the procedure of using the calibration set to determine the threshold for each class. The objective is that the proportion of true labels across prediction sets matches the desired confidence level, as specified by the alpha parameter |
| Calibration set | A calibration set is a labeled subset of the dataset held out from the training set used to estimate the threshold for each class (in Mondrian ICP). |
| Conformal Prediction (CP) | A framework for constructing reliable prediction intervals or sets at a desired confidence level for a given machine learning model. The framework can be used with any machine learning model. |
| Coverage | Coverage of a conformal predictor is the proportion of times that the true label falls within the prediction sets produced by the predictor, over all the inputs. |
| Covariate shift | A phenomenon that occurs when the distribution of the input data changes between the training and testing phases of a machine learning model. It is assumed that the conditional distribution of the target variable given the input features ($P(Y|X)$) remains the same across the training and test sets. |
| Cumulative Distribution Function (CDF) | CDF gives the cumulative probability of the random variable taking on a value less than or equal to a particular value. |
| De novo Drug design model | A machine learning model used to generate novel drug candidates with desired properties. These models are based on generative models such as Variational Autoencoders, Reinforcement Learning, or Genetic algorithm and explore large chemical space. |
| Energy-based model | A type of model that learns a function that assigns low energy scores to data points that are similar to the training data and high energy scores to data points that are dissimilar. |
| Exchangeability | Exchangeability refers to the property of a sequence of random variables such that the joint distribution of any permutation of the variables is the same as the joint distribution of the original sequence. Independent and identically distributed (IID) implies exchangeability. Exchangeability is an important consideration in the Conformal prediction framework. |
| Fingerprint splitting | A method used to divide a dataset of molecules into training and testing sets based on the similarity of their molecular fingerprints. |
| Generative model | A type of machine learning model that learns the distribution of a dataset and can be used to generate new data points (in this case drug molecules) with similar properties. |
| Kernel Density Estimation | Kernel density estimation (KDE) is a non-parametric method for estimating the probability density function of a random variable based on a set of observations. It involves placing a kernel at each data point and summing the kernels to obtain a smoothed estimate of the density function. |
| Mondrian ICP | Mondrian Inductive Conformal Prediction (Mondrian ICP) is a variant of the conformal prediction framework that provides class-wise coverage guarantees for multi-class classification problems. The prediction sets are constructed to provide class-wise coverage guarantees, meaning that they are guaranteed to contain the true class label with a certain probability (determined by a user-defined confidence level) for each class. |

| | |
|---|---|
| Prediction set | A prediction set is a set of candidate labels (class values) for a given input. A prediction set is considered valid if it contains the true class label of an input. |
| Quantile Function | A function that maps a probability to a corresponding value in a distribution. It is the inverse of the cumulative distribution function. |
| Scaffold splitting | A method used to divide a dataset of molecules into training and testing sets while ensuring that the two sets have similar scaffold diversity. |
| Toxicity | Refers to the potential of a drug to cause harm to living organisms. It is an important ADMET property to consider in drug discovery. |
| Validity | A conformal predictor is said to be valid if its coverage level is equal to the user-defined significance level (usually denoted by alpha) used to construct the prediction sets. |

