# OpenReview forum: "CoDrug: Conformal Drug Property Prediction with Density Estimation under Covariate Shift"
_NeurIPS.cc/2023/Conference — NeurIPS 2023 poster_

### Official Review · Reviewer_uhvu · 2023-06-22

**Soundness:** 3 good
**Presentation:** 3 good
**Contribution:** 3 good
**Rating:** 6
**Confidence:** 3

**Summary:**

The paper concerns the problem of conformal drug property prediction, where set valued predictions that should cover with high probability the true class are made. The authors argue that due to sampling design covariate shift tends to occur where test data has different input distribution than the training data, breaking down standard validation set based calibration methods leading to incorrect coverage for predictions. A semi-supervised co-training method and KDE are proposed for obtaining reliable density estimates from validation and test data, in order to re-calibrate the conformal predictions. Theoretical analysis establishes coverage guarantee asymptotically, and experiments performed on drug discovery and drug design data sets show how basic approaches can easily break and that the proposed methods can provide more proper coverage predictions.

**Strengths:**

Drug property prediction applications (as well as many other similar molecular prediction tasks) routinely break basic i.i.d. assumptions, and thus there is clear need for ML methods that remain valid even under such covariate shift. Though closely related works exist, the exact formulation of the problem appears novel, and the proposed solution is both theoretically justified and empirically validated on two real-world data sets. The paper is fairly clearly written, recent literature well covered, and the Figure well illustrates the overall workflow.

**Weaknesses:**

Significance of the work: It is not clear to me, how useful this kind of coverage guarantee would in practice be for researchers working on drug property prediction tasks, and the problems solved in the experiments are somewhat artificial as a split leading to covariate shift is simulated by the authors rather than based on some real world application where it would naturally appear. Contribution would be stronger if the authors could demonstrate the applicability of the approach to more broad range of applications.

**Questions:**

Can you open up a bit the basic motivation of the work: "Furthermore, de novo drug design models or screening datasets sample
molecules from a large chemical space, making the exchangeability assumption invalid." - why and how exactly is this assumption violated in typical study designs?

**Limitations:**

-

---

> ### Author Rebuttal · Authors · 2023-08-10
>
> We thank the author for their overall positive assessment of the paper. In our response, we would like to address the posed questions and elaborate on the stated weaknesses.
>
> ## Clarification regarding the exchangeability argument
> A sequence of random variables is exchangeable if the joint probability distribution remains the same, regardless of the order in which the variables appear. If a sequence of random variables is drawn from i.i.d., then it is also exchangeable because permuting the order does not change the joint probability distribution.  In drug discovery experiments, this assumption is frequently violated because the molecules in the test set do not originate from the same distribution as the training set.  For instance, in various virtual screening and generative drug design models, the primary objective is to generate molecules optimized for a specific target property, like binding affinity or docking score. These screening libraries or generative methods explore an extensive chemical space to search for molecules that exhibit the desired properties.
>
> Consider the task of estimating interval or set properties of generated molecules based on drug properties using an Oracle trained on those attributes (like toxicity or PK/PD).  The distribution of the sampled set of molecules is distinct from the distribution of the molecules seen during training. This is due to the fact that a toxicity model, for example, might have been trained only on molecules with specific structural frameworks or features, violating the assumption of exchangeability.   In the current study, our focus is on developing techniques to produce robust and statistically valid set-valued or interval-valued predictions in such contexts. We use the conformal prediction framework in scenarios where the test and the calibration set are not exchangeable, but linked by covariate shifts.
>
> ## Relevance of data-splitting in real-world
> Although it is challenging to perfectly emulate real-world settings for the application, we have designed experiments using scaffold and fingerprint splitting to mimic distribution shifts. In scaffold splitting, molecules are divided based on Bemis and Murcko scaffolds, effectively reducing a compound to its core components by removing all side chains. This approach closely resembles real-world data collection, where certain biological assays involve molecules with nearly identical scaffolds. Scaffold splitting has become a standard technique in several out-of-distribution (OOD) generalization benchmarks in drug discovery [18] [19]. In addition to scaffold splitting, we also utilize fingerprint splitting, obtained from the DeepChem library [26], to emulate scenarios where the train and test sets are sourced from different regions of the chemical space. This technique allows us to simulate situations commonly encountered in screening library searches and generative models, which explore a vast chemical space. We welcome the reviewer's suggestions for any other approaches to perform splitting with a more realistic distribution shift.
>
> ## Broader applicability of the design setting
> We present a few general examples where the coverage guarantee offered by the conformal prediction framework can be beneficial in drug property prediction:
>
> In classification settings: The Mondrian conformal prediction setting, as demonstrated in this paper, provides class-wise guarantees. This is beneficial in scenarios where certain classes are rare, such as identifying rare active compounds (or non-toxic compounds) in a set of molecules. By setting a target coverage rate, for example, an alpha of 0.1 or a 90\% target coverage, the prediction set will contain the class active (if the original class is active) with a marginal probability of 90\%. This can potentially aid in cost-saving by preventing the omission of molecules with desired properties during experimentation.
>
> In multi-label classification: Consider drug-drug interaction prediction, where, the objective is to generate a set of drug molecules with potential interactions for a given drug molecule or a set of potential interactions for a drug-drug pair. It is crucial to include all drug molecules with possible interactions confidently and the CP framework can help provide prediction sets with a coverage guarantee. This approach can also be applied similarly in drug-adverse reaction prediction tasks.
>
> In regression: In drug discovery, predicting continuous properties like potency, affinity, and solubility is crucial. Conformal prediction in regression tasks provides prediction intervals at a specified coverage rate. For instance, a 90% target coverage rate implies the interval holds true for about 90% of cases. This helps researchers gauge uncertainty, prioritizing experimental efforts on molecules with the highest potential for desired properties.
>
> We believe that the proposed framework can be adapted to a wide range of tasks and settings beyond the small molecule classification tasks discussed in this paper, and our future work will focus on exploring these possibilities.
> This includes settings involving regression, multi-label classification, and utilization in other chemical/biological contexts like RNA sequence/structure analysis, protein property prediction, as well as peptide and antibody design. These tasks frequently encounter covariate shifts, making the adaptability of the proposed framework highly relevant.
>
> [18] Ji, Yuanfeng, et al. "DrugOOD: Out-of-Distribution (OOD) Dataset Curator and Benchmark for AI-aided Drug Discovery--A Focus on Affinity Prediction Problems with Noise Annotations." arXiv preprint arXiv:2201.09637 (2022).
> [19] Gui, Shurui, et al. "Good: A graph out-of-distribution benchmark." Advances in Neural Information Processing Systems 35 (2022): 2059-2073.
> [26] Ramsundar, Bharath, et al. Deep learning for the life sciences. " O'Reilly Media, Inc.", 2019.

---

> > ### Comment · Reviewer_uhvu · 2023-08-14
> >
> > Thank you for the clarifications regarding the exchangeability argument, as well as discussion about the experimental design and broader applicability of the approach.

---

### Official Review · Reviewer_nCYs · 2023-07-06

**Soundness:** 3 good
**Presentation:** 3 good
**Contribution:** 3 good
**Rating:** 7
**Confidence:** 4

**Summary:**

This paper studies conformal drug property prediction under covariate shift, i.e. the calibration data and the test data are not exchangable.
The proposed method, CoDrug, uses an energy-based model trained on both labeled and unlabeled data to model the conditional $p(y|x)$, and KDE to estimate the likelihood ratio between the calibration and test dataset, which is subsequently used to rectify for the distribution shift.

Experiments demonstate CoDrug effectively reduces coverage gap compared to unweighted models. A demo is also given where CoDrug is applied to estimate the logP values of a set of de novo designed molecules.

**Strengths:**

**Originality**

This work addresses uncertainty quantification in molecule property prediction under covariate shift. The authors proposed a novel method to estimate the likelihood ratio, and applied it to the molecular property prediction task.

**Quality**

The overall paper quality is good. Contributions are clearly stated. The math is solid. Most model choices are reasonable or justified. The experimental setting is fair. And a nice, realistic demo is given as a dessert.

**Clarity**

The writing is easy-to-follow in general.

**Significance**

The task is important and realistic in drug discovery.

**Weaknesses:**

**Clarity**

Perhaps Section 4.2 (implementation) could be put after 4.3 (the big picture).
Currently Section 5.3 is not clear enough. At first I thought you are using this model to generate new molecules. Also, I believe this section can be shortened to free up some spaces in the current somewhat overcrowded paper.
If you have the time, figure 1 could use a little bit of improvement.


\* See questions for other points to be clarified.

**Questions:**

1. How important is the unlabeled ZINC dataset to the experimental results?
2. Could you explain, theoretically and/or empirically, why I couldn't use a generative model trained on ZINC to estimate the likelihood ratio?
3. Why is ReLU not applied in the provided code when getting classification features?
```python
class ClassificationFeatureModel(nn.Module):
    def __init__(self, model_config):
        super().__init__()
        # several lines omitted
        self.op_1 = nn.Linear(feature_len, int(feature_len / 2))
        self.op_2 = nn.Linear(int(feature_len/2), 8)
        self.op = nn.Linear(8, self.num_outputs)

    def forward(self, mol_graph_batch):
        node_features = self.gconv(
            mol_graph_batch, mol_graph_batch.ndata["h"], mol_graph_batch.edata["e"]
        )
        gfeat = self.readout(mol_graph_batch, node_features)

        op1 = F.relu(self.op_1(gfeat))
        op2 = F.relu(self.op_2(op1))

        outputs = self.op(op2)
        return outputs

    def get_features(self, mol_graph_batch):
        node_features = self.gconv(
            mol_graph_batch, mol_graph_batch.ndata["h"], mol_graph_batch.edata["e"]
        )
        gfeat = self.readout(mol_graph_batch, node_features)

        op1 = self.op_1(gfeat)
        op2 = self.op_2(op1)
        return op2
```

**Limitations:**

I believe the authors should have adressed more to the limitations of this paper. I suspect this is due to the page limit.
I don't see any significant unreported negative societal impact.

---

> ### Author Rebuttal · Authors · 2023-08-09
>
> We thank the reviewer for their overall positive assessment of the paper. We would like to address the questions/concerns raised by the reviewer.
>
> ## Importance of ZINC unlabelled data
> We report the importance of using ZINC-based unlabelled data in Table 3 and Section 5.2 of our paper. The "CoDrug (NoEnergy)", corresponds to the loss function trained without the energy regularization (i.e without the unlabelled ZINC data) as per Eq. (7). In our experiments, we observe a general improvement in bridging the coverage gap by including unlabelled ZINC dataset in our training procedure. Notably, we observed an average increase of 5\% (at an alpha=0.3) and 4\% (at an alpha=0.2) in coverage when analyzing the worst 25% of the experiments.
>
> ## Estimating generative model trained on ZINC to estimate the likelihood ratio
> We are unsure of the reviewer's intended meaning when mentioning "using a generative model trained on ZINC to estimate likelihood ratio."  One way is to utilize features extracted from a generative model trained on ZINC, such as a VAE. In that case, it is indeed feasible to extract features from the generative model. However, a challenge here is that features often have hundreds of dimensions, making KDE more data-hungry, and requiring a larger number of data points for accurate density estimation. Nevertheless, it is still possible to use these features as inputs to train the EBM and extract low-dimensional features, which might yield similar results as our approach.
>
> Another potential way is to use a generative model trained on ZINC data to directly compute the likelihood of samples.  That's an interesting direction to explore, which may be suitable for larger datasets.
> This requires fine-tuning a generative model like VAE or normalizing flow for each specific dataset/task distribution, for both the test and calibration datasets. Learning this distribution can be challenging when the calibration and test sets consists of only a few hundred data points or lesser.
>
>
> ## ReLU not applied in the provided code
> We apologize to the reviewer for the confusion. In our experiments, we utilized features extracted from the "EnergyFeaturesModel," which is also available in the same model.py file. This class has the ReLU function applied while extracting classification features. We will eliminate any such references when we release the final code version.
>
> ## Regarding suggestions for improving the clarity of the paper
> We thank the reviewer for their suggestions on enhancing the paper's clarity and understanding of the limitations imposed by space constraints.
> In light of the suggestions provided, we will reorganize the paper by placing section 4.3 before section 4.2, and we will also make necessary adjustments to improve the coherence of the text. Additionally, in order to eliminate any potential ambiguity in section 5.3, we will carefully review the content to enhance its clarity. We will explicitly state that the molecules being tested originate from a distinct black box generative model, and our current method serves the purpose of providing uncertainty estimates on this set of molecules.
>
> We also appreciate the reviewer's understanding of the challenge imposed by the space constraints. We have outlined the general limitations in the conclusion section and specified the limitations concerning KDE in the experiments section. We will make necessary revisions to emphasize the limitations more prominently in the text for the final version.

---

> > ### Comment · Reviewer_nCYs · 2023-08-14
> > **Reply to the authors**
> >
> > Thank you for the clarifications.

---

### Official Review · Reviewer_e8Bh · 2023-07-07

**Soundness:** 3 good
**Presentation:** 3 good
**Contribution:** 2 fair
**Rating:** 3
**Confidence:** 3

**Summary:**

The paper is concerned with the problem of conformal drug property prediction by formulating and solving the problem as a conformal prediction problem under covariate shift. To this end, the paper proposed training an energy-based model using both labeled and unlabeled data.  To derive conformal prediction under covariate shift, the paper makes use of kernel density estimation to estimate the covariate distribution in training time and in test time. The paper then proceeds to use the estimated density to counteract covariates shift. Theoretical guarantees are given about the proposed methods. Experiments are conducted on four drug toxicity datasets and de novo drug development experiments. The proposed method is compared to a baseline in the experiments.

**Strengths:**

* The paper is concerned with an important practical problem in drug design. It provides a good application of conformal drug prediction under covariate shift.
* Experiments are conducted on a wide variety of datasets that are relevant to the problem of drug design.
* The presentation of the paper is clear.

**Weaknesses:**

* One of my major concerns is the originality of the paper. While the paper makes reasonable choices to develop the proposed method to deal with conformal prediction under covariate shifts for conformal drug properties predictions, many of these choices appeared to be straightforward applications of existing works. It is not clear to me whether the original contribution of the paper is sufficient for acceptance.

* In terms of empirical evaluation, while I think the methods are compared on a fair amount of various datasets related to drug design, I am not quite sure if the proposed method has been compared to enough alternative methods to understand the effectiveness of the proposed method. As such, the paper can benefit from a more exhaustive comparison of related work in the literature.

* While there are theoretical justifications mentioned in the paper. They are mostly straightforward derivations from existing work. Although this is fine, it does not seem to contribute to the strength of the paper.

**Questions:**

* The authors may further explain why the proposed method has sufficient novelty.
* The authors can also consider introducing more alternative methods for comparison empirically.

**Limitations:**

The authors mention the limitation of their work in the conclusion section.

---

> ### Author Rebuttal · Authors · 2023-08-09
>
> We thank the reviewer for their feedback and would like to address the concerns regarding the novelty of our paper. Our work presents several contributions, which we believe are novel for various reasons:
>
> - We focus on conformal prediction (CP), a powerful and increasingly popular tool for obtaining distribution-free uncertainties. While CP has been applied to high-stakes applications including drug discovery ([8] [9] [10]), we identify and tackle the challenge of providing statistically rigorous uncertainty quantification in the presence of distribution shift—a common occurrence in drug discovery due to the vastness of the chemical space. Previous studies primarily concentrated on non-conformity measures, neglecting the critical consideration of distribution shift, which is a key aspect that significantly impacts practical usability.
> - To this end, we first show that the coverage guarantee offered by the conformal prediction framework is indeed violated in a practical drug discovery context. We devise experiments that simulate realistic drug discovery scenarios, employing fingerprint and scaffold splitting techniques. The experiments clearly demonstrate the violation of the CP coverage guarantee and shed light on the existing methods' limitations. Leveraging recent advances in CP under covariate shifts, we propose and construct a framework that effectively bridges the coverage gap. This framework is showcased through experiments on multiple property prediction datasets, showcasing its applicability in real-world scenarios.
> - We tackle the challenge of obtaining likelihood ratios in CP under covariate shift, which is not immediately clear from prior work ([11]). We propose a novel approach leveraging Kernel Density estimates to correct for covariates, and we empirically demonstrate its effectiveness. Further,  we show that the kernel density estimates are consistent, which means that the covariate shift is precisely adjusted for, and the coverage guarantee is recovered asymptotically.
> - We also develop a training framework that utilizes unlabeled data and an energy-based model formulation. This framework facilitates in extracting features that enable accurate computation of likelihoods using Kernel Density Estimation (KDE). Our results demonstrate the effectiveness of this end-to-end approach in addressing the coverage gap.
> -  In AI-based drug discovery, there has been a substantial focus on developing molecule-generative models ([3]-[6]). Despite their potential, their practicality is limited by the fact that only a small number of these molecules can be manufactured and experimentally tested. Thus, precise property estimates, accompanied by robust uncertainties, are crucial. We demonstrate that our proposed framework can be used with the molecules sampled from generative models, providing them with the ability to quantify uncertainties in a statistically robust manner.
>
> ## Alternative methods for empirical comparisons:
> We believe that we made our best efforts to conduct a comprehensive benchmark of the proposed method. In our experiments, we perform:
> 1. Comparisons with vanilla CP without any correction, which is the most common approach used in various methods.
> 2. We conducted ablation studies on different components of our approach - feature extraction and the energy-based training procedure.
> 3. We also compared our method by using the likelihood estimation approach suggested by [11].
>
> Considering the novel nature of uncertainty estimation for drug discovery under distribution shift, we welcome any reviewer's insights and suggestions for alternative methods of comparison that they may have in mind.
>
> [3] Fu, Tianfan, et al. "Reinforced genetic algorithm for structure-based drug design." Advances in Neural Information Processing Systems 35 (2022): 12325-12338.
> [4] You, Jiaxuan, et al. "Graph convolutional policy network for goal-directed molecular graph generation." Advances in neural information processing systems 31 (2018).
> [5] Gómez-Bombarelli, Rafael, et al. "Automatic chemical design using a data-driven continuous representation of molecules." ACS central science 4.2 (2018): 268-276.
> [6] Jin, Wengong. et. al. Junction tree variational autoencoder for molecular graph generation. In ICML, pages 2323–2332.PMLR, 2018.
> [8] Zhang, Jin., et al. (2021). Deep learning-based conformal prediction of toxicity. Journal of chemical information and modeling, 61(6), 2648-2657.
> [9] Isidro Cortés-Ciriano and Andreas Bender. Concepts and applications of conformal prediction in computational drug discovery. arXiv preprint arXiv:1908.03569, 2019.
> [10] Isidro Cortés-Ciriano and Andreas Bender. Deep confidence: a computationally efficient framework for calculating reliable prediction errors for deep neural networks. Journal of chemical information and modeling, 59(3):1269–1281, 2018.
> [11] Tibshirani, Ryan J., et al. "Conformal prediction under covariate shift." Advances in neural information processing systems 32 (2019).

---

> > ### Comment · Reviewer_e8Bh · 2023-08-20
> > **Thank you for the additional explantion!**
> >
> > Thank you for providing additional explanation regarding my questions. I am less concerned about whether the alternative methods are sufficient based on these additional contexts.  The authors may consider adding this explanation to the paper to justify their method selection.  In terms of novelty, the proposed method addresses a more challenging scenario compared to existing work such as [11], but I am not sure if the proposed method is sufficiently original.

---

> > > ### Author Response · Authors · 2023-08-20
> > >
> > > Thank you for your response. In accordance with the suggestions, we will make the necessary edits to the manuscript to incorporate additional explanations regarding our choice of comparisons. We would also like to thank the reviewer for acknowledging that the problem being addressed in the current work presents a more challenging scenario compared to existing works.

---

### Official Review · Reviewer_SDHG · 2023-07-07

**Soundness:** 3 good
**Presentation:** 4 excellent
**Contribution:** 2 fair
**Rating:** 7
**Confidence:** 4

**Summary:**

The authors apply conformal prediction under covariate shift to molecular property prediction. They propose using KDE to estimate the densities for the calibration and test sets, and demonstrate that doing so is better than plain vanilla CP.

**Strengths:**

### Originality

This paper puts together existing ideas that provide uncertainty estimates for molecular property prediction and related tasks.

### Quality

Quality is high, the method is evaluated on many different prediction and generative tasks.

### Clarity

The paper is well-written and easy to understand.

### Significance

Proper uncertainty estimation is important for many molecular prediction tasks, and this paper provides a CP method with guarantees that deals with the central issue of such tasks, namely covariate shift.

**Weaknesses:**

- Should show under what conditions using KDE fails.

**Questions:**

No questions

**Limitations:**

There is no discussion of limitations.

---

> ### Author Rebuttal · Authors · 2023-08-09
>
> We thank the reviewer for the overall positive assessment of the paper. In our response, we would like to address the question about the applicability of KDE and elaborate on the limitations of the proposed framework.
>
> - While we demonstrated that KDE can provide asymptotic coverage guarantees, this may not necessarily translate to improved performance in scenarios with limited sample sizes. Although our experiments demonstrate that the convergence tends to approach the target value in most scenarios, we do acknowledge that there are a few instances where the improvement in coverage is limited. For example, in Table 2, we observe that the improvement in coverage for ClinTox (Y=1) is limited due to the relatively small size of the calibration set, consisting of only 19 points. As such, an important area for further research is to explore ways to obtain likelihood ratios that are more data-efficient, particularly in scenarios with smaller calibration sets.
>
> - It is worth noting that our current work focuses on addressing the coverage gap in classification tasks, and regression tasks were not included in this study. However, we recognize the importance of uncertainty quantification in regression settings, especially for various critical drug properties represented as regression problems. As part of our future work, we intend to explore and extend the proposed framework to encompass regression tasks.
>
> - Additionally, our current work primarily focuses on small molecules. Nevertheless, we recognize that covariate shift is a common occurrence in various chemical and biological phenomena, making the framework presented in this work applicable in broader contexts. Nonetheless, obtaining high-quality feature vectors for computing likelihood for different applications remains a challenge that warrants further research.
>
> In our current version of the paper, we have outlined the general limitations in the conclusion section and specified the limitations concerning KDE in the experiments section. We will make necessary revisions to emphasize the limitations more prominently in the text.

---

> > ### Comment · Reviewer_SDHG · 2023-08-21
> >
> > Many thanks to the authors for their response and addressing my review. I have raised my rating to Accept.

---

### Comment · Area_Chair_b6eb · 2023-08-18
**Rebuttal**

Thank you for your rebuttal. We will take it into account in making the final recommendation.

---

### Decision · Program_Chairs · 2023-09-21

**Decision:**

Accept (poster)

**Comment:**

The paper puts forward a method that allows accurate conformal prediction of drug properties in the case of covariate shift, that is, when the test distribution does not match the training distribution, which often is the case in practice. The authors analyze the method theoretically proving asymptotic convergence and evaluate its behaviour empirically. The reviewers find the paper well written and technically sound. The reviewers posed some questions on the significance of the contribution, which were addressed by the rebuttal. On balance, the contribution appears to be sufficient.